# Proteomic identification of biomarkers in maternal plasma that predict the outcome of rescue cerclage for cervical insufficiency

**Kisoon Dan**[1,☯]**, Ji Eun Lee**[2,☯]**, Dohyun Han**[1]**, Sun Min Kim**[3]**, Subeen Hong**[4]**, Hyeon Ji Kim**[5]**, Kyo Hoon Park**[5]***

**1** Proteomics Core Facility, Biomedical Research Institute, Seoul National University Hospital, Seoul, Korea, **2** Biomedical Research Division, Theragnosis Research Center, Korea Institute of Science and Technology, Seoul, Korea, **3** Department of Obstetrics and Gynecology, Seoul Metropolitan Government Seoul National University Boramae Medical Center, Seoul, Korea, **4** Department of Obstetrics and Gynecology, College of Medicine, The Catholic University of Korea, Seoul, Korea, **5** Department of Obstetrics and Gynecology, Seoul National University College of Medicine, Seoul National University Bundang Hospital, Seongnam, Korea

☯ These authors contributed equally to this work.

* pkh0419@snubh.org

**Data Availability Statement:** All relevant data are within the paper and its Supporting Information files.

## Abstract

### Objective

We sought to identify plasma protein biomarkers that are predictive of the outcome of rescue cerclage in patients with cervical insufficiency.

### Methods

This retrospective cohort study included 39 singleton pregnant women undergoing rescue cerclage for cervical insufficiency (17–25 weeks) who gave plasma samples. Three sets of pooled plasma samples from controls (cerclage success, n = 10) and cases (cerclage failure, n = 10, defined as spontaneous preterm delivery at <33 weeks) were labeled with 6-plex tandem mass tag (TMT) reagents and analyzed by liquid chromatography-tandem mass spectrometry. Differentially expressed proteins between the two groups were selected from the TMT-based quantitative analysis. Multiple reaction monitoring-mass spectrometry (MRM-MS) analysis was further used to verify the candidate proteins of interest in patients with cervical insufficiency in the final cohort (n = 39).

### Results

From MRM-MS analysis of the 40 proteins showing statistically significant changes ($P <$ 0.05) from the TMT-based quantitative analysis, plasma IGFBP-2, PSG4, and PGLYRP2 levels were found to be significantly increased, whereas plasma MET and LXN levels were significantly decreased in women with cerclage failure. Of these, IGFBP-2, PSG4, and LXN levels in plasma were independent of cervical dilatation. A multiple-biomarker panel was developed for the prediction of cerclage failure, using a stepwise regression procedure, which included the plasma IGFBP-2, PSG4, and LXN (area under the curve [AUC] = 0.916).

**Funding:** KHP received two grants as described below. This study was supported by the Seoul National University Bundang Hospital Research Fund (Grant No. 13-2016-006), and by the National Research Foundation of Korea (NRF) grant funded by the Korea government (MSIT) (No. 2020R1F1A1048362). The funders had no role in study design, data collection and analysis, decision to publish, or preparation of the manuscript.

**Competing interests:** The authors have declared that no competing interests exist.

The AUC for this multiple-biomarker panel was significantly greater than the AUC for any single biomarker included in the multi-biomarker model.

## Conclusions

Proteomic analysis identified useful and independent plasma biomarkers (IGFBP-2, PSG4, and LXN; verified by MRM) that predict poor pregnancy outcome following rescue cerclage. Their combined analysis in a multi-biomarker panel significantly improved predictability.

## Introduction

Although the incidence of cervical insufficiency is relatively rare, occurring in <0.5% of all pregnancies, it is one of the main causes of mid-trimester spontaneous abortion and/or early spontaneous preterm birth [1–3]. An emergency (or rescue) cerclage is currently the only method to prolong pregnancy and salvage the fetus in women presenting with a dilated cervix and/or prolapsed membranes in the second trimester. In such cases, this procedure results in neonatal survival rates of 72% [4]. However, little is known about the biomarkers, especially in non-invasive samples, that can predict the success of rescue cerclage in women with cervical insufficiency and help identify ideal candidates for cerclage placement.

Traditionally, the most accurate biomarker-based method to predict clinical success in women undergoing rescue cerclage has been to analyze amniotic fluid (AF) samples, obtained by abdominal amniocentesis, for infectious/inflammatory status and decidual hemorrhage [5–9]. Recently, our group also reported several biomarkers in the AF which can be used to predict the outcome of rescue cerclage using proteomic-based approaches [9]. However, second trimester amniocentesis is invasive and may pose the risk of membrane rupture, thus limiting its clinical utility [10]. Importantly, several studies have reported significant changes in various proteins, occurring simultaneously in the AF and maternal blood compartments, in the setting of decidual hemorrhage and microbial invasion of amniotic cavity [11–13]. Therefore, a maternal blood sample may be a feasible alternative to AF samples. However, to date, there is little information on the role of multiple protein mediators in the maternal blood, especially when evaluated using a high-throughput approach, in predicting adverse outcomes in women undergoing emergency cerclage for cervical insufficiency.

Of note, recently, applications, such as mass spectrometry (MS)-based proteomic techniques, have been shown to be useful in the discovery of novel protein biomarkers associated with complex conditions with multiple causes, including spontaneous preterm delivery (SPTD) [14, 15]. Specifically, in the setting of preterm labor, novel markers of SPTD have been reported using proteomic analysis of serum samples [16–18]. However, to date, there has been no study using this approach to identify plasma biomarkers of pregnancy outcome after rescue cerclage for cervical insufficiency. Therefore, we aimed to comprehensively identify plasma protein biomarkers that are predictive of the outcome of rescue cerclage in patients with acute cervical insufficiency, using tandem mass tag (TMT)-based liquid chromatography-tandem mass spectrometry (LC-MS/MS), followed by multiple reaction monitoring-mass spectrometry (MRM-MS) analysis.

## Materials and methods

### Study design

This study was approved by the ethics committee at Seoul National University Bundang Hospital (IRB no. B-1311/228-010). Written informed consent was obtained from all participants for the collection and use of blood samples for research purposes. All patients were recruited at the Seoul National University Bundang Hospital (Seongnamsi, Republic of Korea) between September 2004 and December 2015. The study population consisted of women with a singleton pregnancy at 17 to 25 weeks of gestation, who underwent rescue cerclage after diagnosis of acute cervical insufficiency. The inclusion criteria were as follows: (1) a live fetus, (2) intact amniotic membranes, and (3) the availability of an aliquot of a maternal plasma sample for analysis. Women with multiple pregnancies, major congenital anomalies, clinical chorioamnionitis at presentation, or prophylactic cerclage during early pregnancy were excluded from the study. A total of 39 women undergoing rescue cerclage for cervical insufficiency were enrolled in the study. Cervical insufficiency was defined as a painless spontaneous dilatation of the cervix $\geq$ 1 cm on physical examination, associated with exposed fetal membranes, as determined by visual assessment during a sterile speculum examination, without any uterine contractions.

We performed a nested case-control study for biomarker discovery using stored maternal plasma samples from 10 case patients who had subsequent SPTD at < 33 weeks of gestation after cerclage placement and 10 control patients who delivered at $\geq$ 33 weeks. Case patients were randomly selected from a subgroup of 23 women with SPTD at < 33 weeks from a total cohort of 39 women who had undergone rescue cerclage for cervical insufficiency and who met the inclusion and exclusion criteria described above. Each control patient who had rescue cerclage for cervical insufficiency was matched for gestational age at sampling, cervical dilatation, parity, years of cerclage placement, and maternal age with a case patient. The proteomic profiles of maternal plasma samples were compared between the case and control groups using TMT-based quantitative analysis. To verify the biomarker candidates selected from the discovery experiment, MRM-MS was performed in the final cohort of 39 individual samples.

### Management of cervical insufficiency and collection and storage of plasma samples

Women with acute cervical insufficiency were scheduled to receive rescue cerclage using the McDonald technique under spinal anesthesia. For women with advanced cervical dilatation and bulging of fetal membranes, amnioreduction and an inflated Foley catheter were used to decrease intra-amniotic fluid pressure and replace the prolapsed fetal membranes. The use and type of antibiotic and tocolytic agent (ritodrine, magnesium sulfate, or atosiban) were determined by the attending physician. Antenatal corticosteroids for enhanced fetal lung maturation were administered to women at 24 + 0 to 33 + 6 weeks of gestation. A more detailed description of the method for rescue cerclage and the medication given to patients undergoing these procedures have been published elsewhere [6].

Before cerclage placement, maternal blood samples were obtained by venipuncture and collected into ethylenediaminetetraacetic acid tubes. White blood cell counts and C-reactive protein concentrations in the maternal blood samples were measured. Blood samples were centrifuged at 1,500 ×$g$ for 10 min, after which the supernatant was aliquoted and stored frozen at -80˚C until future use. Plasma samples with significant hemolysis were excluded from the study.

### TMT-based proteome profiling, followed by verification of candidate proteins using MRM-MS

From the controls (n = 10) and cases (n = 10) used in the discovery experiments, sets of three, three, and four plasma samples were pooled with equal amounts of each sample, resulting in three sets of pooled samples from both the control and case groups. This pooling strategy (three pools, containing three, three, or four samples each) has merit in terms of (1) reducing the disadvantage inherent in pooling all 10 samples from each group together, and (2) providing three independent biological replicates in each group that can be used for six-plex TMT labeling-based quantitative analyses. These pooled plasma samples were subjected to immuno-depletion, tryptic digestion, labeling with TMT tags of the peptides produced from tryptic digestion, high-pH reversed-phase peptide fractionation, LC-ESI-MS/MS analysis, data processing for protein identification and quantification, and bioinformatics analysis to identify differentially expressed proteins (DEPs) between the control and case groups. The DEPs were then subjected to gene ontology (GO) analysis for functional classification and further validated using MRM-MS analysis (Fig 1; see details in the S3 File). GO analysis was performed using the DAVID bioinformatics tool (http://david.abcc.ncifcrif.gov/).

### Statistical analysis

Clinical data and abundance levels of candidate proteins were compared using a Mann-Whitney U-test for continuous non-parametric data and a $\chi^2$-test or Fisher's exact test for categorical data. MRM results were analyzed using a multivariate logistic regression model to examine the independent relationships between candidate biomarker levels in the plasma and the occurrence of SPTD at < 33 weeks, after controlling for baseline clinical variables (i.e., cervical dilatation), with a $p$-value <0.05 during univariate analysis. In the logistic regression model, continuous data for various proteins were transformed into dichotomous data for prediction or decision-making purposes. Receiver operating characteristic (ROC) curves were created for each candidate protein and used to determine the optimal cut-off values (defined using the maximum Youden index (maximum [sensitivity + specificity–1])) for dichotomization. Using a previously described method [19], we calculated and compared the areas under the ROC curves (AUCs) for each protein. Finally, to determine the best protein panel, based on candidate plasma biomarkers, to predict the outcome of rescue cerclage for cervical insufficiency, a multivariate logistic regression analysis was performed using the backward stepwise method. A Kaplan-Meier survival curve was used to analyze the interval from cerclage to delivery, and log-rank tests were performed to evaluate differences in the cerclage-to-delivery interval between the two curves. The data were fitted to Cox proportional hazards models for multivariate analysis, adjusting for advanced cervical dilatation. All probability values are 2-tailed, and $P$ values < 0.05 were considered to be statistically significant. The data analyses were performed with SPSS version 25.0 (IBM SPSS Inc., Chicago, IL).

## Results

### Baseline characteristics of the discovery cohorts

The baseline characteristics of the exploratory cohorts used for TMT-based quantitative proteomic analysis are presented in S1 Table. Because matching was performed, the patients of the case and control groups were similar with respect to advanced cervical dilatation at presentation, gestational age at sampling, maternal age, or parity.

## A. Discovery phase          B. Verification phase

**Fig 1. Schematic workflow of the discovery (TMT labeling-based quantification) and verification (LC-MRM MS) experiments.**
SPTD, spontaneous preterm delivery; HPLC, high-performance liquid chromatography; LC-MS/MS, liquid chromatography-tandem
mass spectrometry; SIS, stable isotope-labeled standard.

### Experimental design for biomarker discovery and verification using proteomic analysis

Fig 1 describes the general workflow for the discovery and verification of plasma biomarkers
to predict the pregnancy outcome after rescue cerclage. MS analysis of the TMT-labeled sam-
ples identified 818 proteins and 777 quantifiable proteins with high confidence, at a 1% false
discovery rate (S2 Table). TMT quantification, based on MS2 reporter ion intensity, identified
40 DEPs with a *P*-value < 0.05 (S3 Table). Thirty-three proteins were up-regulated in the case
group and seven proteins were up-regulated in the control group (S1 Fig).

### Gene ontology enrichment analysis of the identified DEPs

To functionally classify proteins showing statistically significant changes between control and
case groups, we performed GO enrichment analysis using the DAVID database (S2 Fig). The
top five enriched biological processes of the DEPs were locomotion, regulation of cell migra-
tion, proteolysis, cell migration, and peptidyl-tyrosine phosphorylation. With regard to

molecular function of the plasma proteins enriched in SPTD cases, the top five molecular functions in order of significance were: glycosaminoglycan binding, heparin binding, sulfur compound binding, peptidase activity, and receptor binding. Moreover, with regard to the cellular component of the GO analysis, most of the identified DEPs were classified as extracellular proteins.

## Verification of differentially expressed proteins using MRM-MS

To verify the 40 DEPs found in the TMT-based quantitative analysis, a targeted multiplexed peptide MRM-MS assay was used to analyze 39 individual plasma samples. After optimization of the MRM methods (S3 File and S3 Fig and S4 and S5 Tables), we measured the relative abundance of 59 surrogate peptides from the 37 DEPs in individual plasma samples. After performing a Mann-Whitney U test to compare the light (endogenous peptide) to heavy peptide peak area ratio (PAR) of each peptide, seven peptides exhibited statistically significant changes ($P < 0.05$). The levels of LEGEACGVYTPR (insulin-like growth factor-binding protein 2 [IGFBP-2]), LIQGAPTIR (IGFBP-2), DVLTFTCEPK (pregnancy specific beta-1-glycoprotein 4 [PSG4]), IIYGPAYSGR (PSG4), and TDCPGDALFDLLR (peptidoglycan recognition proteins 2 [PGLYRP2]) were found to be significantly higher in women who had SPTD at < 33 weeks, than in women in the control group (Table 1 and Fig 2). In contrast, the levels of GDLTIANLGTSEGR (hepatocyte growth factor receptor [MET]) and FAVEEIIQK (latexin [LXN]) were significantly lower in women with SPTD at < 33 weeks (Table 1 and Fig 2).

Using ROC analyses of the PARs, we further assessed the potential of the seven most markedly dysregulated peptides for prediction of SPTD prior to 33 weeks of gestation after cerclage (Table 1). The AUCs for the seven peptides ranged from 0.686 to 0.764 and **did not** significantly differ from each other (all factors: $P = 0.22–1.00$).

The mean cerclage-to-delivery interval was $55.72 \pm 43.74$ days (range, 2–141 days). Unlike the results analyzed in the discovery cohort, univariate analysis in the final cohort showed a significant association between cervical dilatation and the occurrence of SPTD at < 33 weeks (Table 2). Thus, we adjusted for baseline risk factors, such as cervical dilation, in multivariate analyses. In a multiple logistic regression model, continuous factors were entered as dichotomous covariates using the cut-off points obtained from the ROC

**Table 1. The relative abundance of plasma biomarkers in relation to the occurrence of spontaneous preterm delivery at < 33 weeks after cerclage, areas under the curves, and optimal cut-off values for every protein.**

| Peptide sequence | Delivery <33 weeks | Delivery ≥33 weeks | P-value | AUC | Cutoff value | Sensitivity[a] | Specificity[a] |
|---|---|---|---|---|---|---|---|
| | (n = 23) | (n = 16) | | | | | |
| LEGEACGVYTPR (IGFBP-2) | 0.115 (0.065–0.231) | 0.089 (0.070–0.153) | 0.007 | 0.758 | ≥0.098 | 78.3 | 75.0 |
| LIQGAPTIR (IGFBP-2) | 0.083 (0.048–0.211) | 0.064 (0.049–0.119) | 0.050 | 0.686 | ≥0.072 | 65.2 | 68.7 |
| DVLTFTCEPK (PSG4) | 0.0528 (0.013–0.285) | 0.022 (0.006–0.304) | 0.040 | 0.696 | ≥0.035 | 78.3 | 62.5 |
| IIYGPAYSGR (PSG4) | 0.347 (0.057–1.078) | 0.113 (0.039–1.600) | 0.018 | 0.726 | ≥0.199 | 82.6 | 69.7 |
| TDCPGDALFDLLR (PGLYRP2) | 0.980 (0.566–1.945) | 0.729 (0.407–1.293) | 0.049 | 0.687 | ≥0.785 | 78.3 | 56.2 |
| GDLTIANLGTSEGR (MET) | 0.037 (0.021–0.063) | 0.044 (0.032–0.098) | 0.027 | 0.711 | ≤0.040 | 65.2 | 68.7 |
| FAVEEIIQK (LXN) | 0.463 (0.208–0.938) | 0.602 (0.323–0.754) | 0.006 | 0.764 | ≤0.512 | 73.9 | 81.2 |

AUC, areas under the curves; IGFBP-2, insulin-like growth factor-binding protein 2; PSG4, pregnancy specific beta-1-glycoprotein 4; PGLYRP2, peptidoglycan recognition proteins 2; MET, hepatocyte growth factor receptor; LXN, latexin.

Data are given as the median (range) (Peak area ratio).

[a] Values are given as %.

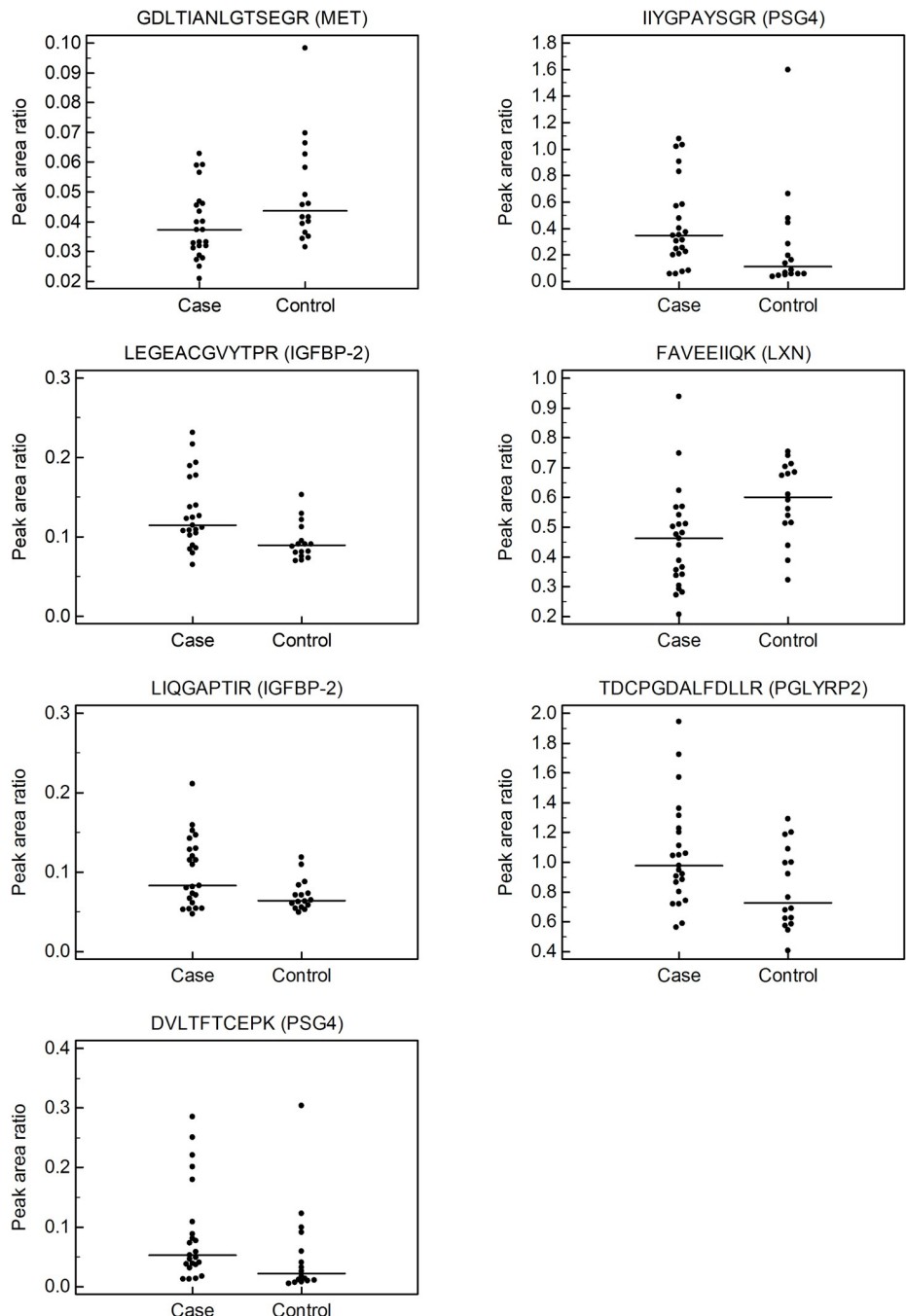

**Fig 2. Interactive plots for significantly differentially expressed (3 up-regulated and 2 down-regulated) proteins in plasma of women with spontaneous preterm delivery (SPTD) at < 33 weeks of gestation after cerclage placement (case group), as determined by MRM assay.** Interactive plots were generated using the normalized peak area of each MRM target peptide. The horizontal line in each figure indicates the median value. GDLTIANLGTSEGR (hepatocyte growth factor receptor [MET]); IIYGPAYSGR (pregnancy specific beta-1-glycoprotein 4 [PSG4]); LEGEACGVYTPR (insulin-like growth factor-binding protein 2 [IGFBP-2]), FAVEEIIQK (latexin [LXN]); LIQGAPTIR (IGFBP-2); TDCPGDALFDLLR (peptidoglycan recognition proteins 2 [PGLYRP2]); DVLTFTCEPK (PSG4).

curves. The optimal cut-off points for cervical dilatation and the PARs of LEGEACG-VYTPR, LIQGAPTIR, DVLTFTCEPK, IIYGPAYSGR, TDCPGDALFDLLR,

**Table 2. Demographic and clinical characteristics of women involved in the final cohort.**

| | Delivery at <33 weeks (n = 23) | Delivery at ≥33 weeks (n = 16) | *P*-value |
|---|---|---|---|
| Age (years) | 31.0 (27.0–39.0) | 33.0 (24.0–38.0) | 0.295 |
| Nulliparity | 60.9% (14/23) | 50% (8/16) | 0.501 |
| Gestational age at sampling (weeks) | 22.0 (17.3–25.1) | 22.3 (20.0–25.4) | 0.219 |
| Cervical dilatation (cm) | 3.0 (0.5–5.0) | 1.5 (0.5–4.0) | 0.001 |
| ≥3 cm | 65.2% (15/23) | 12.5% (2/16) | 0.001 |
| <3 cm | 34.8% (8/23) | 87.5% (14/16) | |
| Serum C-reactive protein (mg/L) | 3.3 (0.5–33.9) | 5.2 (0.1–32.0) | 0.361 |
| White blood cells count (×10³/mm³) | 11.0 (6.5–18.6) | 10.5 (7.1–13.3) | 0.278 |
| Use of tocolytics | 69.6% (16/23) | 50.0% (8/16) | 0.217 |
| Use of corticosteroids | 47.8% (11/23) | 18.8% (3/16) | 0.093 |
| Use of antibiotics | 100.0% (23/23) | 100.0% (16/16) | |
| Gestational age at delivery (weeks) | 25.1 (19.4–32.3) | 36.8 (33.5–40.5) | <0.001 |

Values are given as median (range) or % (n/N).

GDLTIANLGTSEGR, and FAVEEIIQK (IGFBP-2, IGFBP-2, PSG4, PSG4, PGLYRP2, MET, and LXN, respectively) were ≥ 3 cm, ≥ 0.098, ≥ 0.072, ≥ 0.035, ≥ 0.199, ≤ 0.040, and ≤ 0.512, respectively (Table 1). High PARs for LEGEACGVYTPR, DVLTFTCEPK, and IIYGPAYSGR and a low PAR for FAVEEIIQK were significantly associated with SPTD at < 33 weeks of gestation after cerclage, after adjustment for advanced cervical dilatation (≥ 3 cm, Table 3). However, three candidate biomarker peptides, LIQGAPTIR, TDCPGDALFDLLR, and GDLTIANLGTSEGR were not found to be associated with SPTD at < 33 weeks, after adjustment for advanced cervical dilatation (Table 3).

**Table 3. Multivariable logistic regression model showing the adjusted odds ratios of association between various proteins in maternal plasma and spontaneous preterm delivery at <33 weeks after adjusting for advanced cervical dilatation (≥ 3cm).**

| Variables[a] | Adjusted odds ratio (95% confidence interval)[b] | *P*-value[c] |
|---|---|---|
| LEGEACGVYTPR (IGFBP-2) | 8.151 (1.528–43.481) | 0.014 |
| LIQGAPTIR (IGFBP-2) | 3.415 (0.718–16.247) | 0.123 |
| DVLTFTCEPK (PSG4) | 14.852 (1.597–138.134) | 0.018 |
| IIYGPAYSGR (PSG4) | 12.056 (1.948–74.626) | 0.007 |
| TDCPGDALFDLLR (PGLYRP2) | 4.000 (0.789–20.285) | 0.094 |
| GDLTIANLGTSEGR (MET) | 4.768 (0.933–24.359) | 0.061 |
| FAVEEIIQK (LXN) | 15.193 (2.263–101.992) | 0.005 |

IGFBP-2, insulin-like growth factor-binding protein 2; PSG4, pregnancy specific beta-1-glycoprotein 4; PGLYRP2, peptidoglycan recognition proteins 2; MET, hepatocyte growth factor receptor; LXN, latexin.

[a] All continuous predictors were entered as dichotomous variables using the cut-off values derived from the receiver-operating characteristic curves to predict SPTD at <33 weeks.

[a] Variables were dichotomized: high LEGEACGVYTPR (≥ 0.098 vs. < 0.098), high LIQGAPTIR (≥ 0.072 vs. < 0.072), high DVLTFTCEPK (≥ 0.035 vs. < 0.035), high IIYGPAYSGR (≥ 0.199 vs. < 0.199), high TDCPGDALFDLLR (≥ 0.785 vs. < 0.785), low GDLTIANLGTSEGR (≤ 0.040 vs. > 0.040), and low FAVEEIIQK (≤ 0.512 vs. > 0.512).

[b] Adjusted for cervical dilatation ≥ 3cm.

[c] For the adjusted odds ratio.

## Multiple-biomarker panel as an independent predictor of the outcome of rescue cerclage

To develop the optimal multiple-biomarker panel, based on the combination of biomarker candidates, multivariate analysis with a backward selection was carried out on seven bio-marker candidates found to be significant during univariate analysis ($P < 0.05$). For this multi-biomarker combination, all continuous biomarker data were converted to dichotomous covariates using cut-off points derived from the ROC analyses, as detailed in Table 1. A three-protein panel consisting of high PARs for LEGEACGVYTPR (IGFBP-2, $\geq 0.098$) and DVLTFTCEPK (PSG4, $\geq 0.035$) and a low PAR for FAVEEIIQK (LXN, $\leq 0.512$) was identified as the best multiple-biomarker combination (Table 4). The AUC of this multiple-bio-marker panel was 0.916 (95% confidence interval [CI], 0.828–1.004) and the Hosmer-Lemeshow test showed no statistical significance (P = 0.901), suggesting adequate fit to the model. A cut-off point of $\geq 0.53$ was identified as the optimal threshold value for predicting SPTD prior to 33 weeks after rescue cerclage, with a sensitivity of 87.0% (95% CI, 66.4%–97.2%) and a specificity of 81.2% (95% CI, 54.3%–95.9%). The AUC for this multiple-bio-marker panel was significantly greater than the AUC for any single biomarker included in the multi-biomarker model ($P < 0.05$ for each, Fig 3).

## Plasma protein markers and the cerclage-to-delivery interval

Kaplan-Meier survival analyses showed that patients with higher plasma levels of LEGEACG-VYTPR (IGFBP-2) ($\geq$0.098; log-rank test, P = 0.038), DVLTFTCEPK (PSG4) ($\geq$ 0.035; log-rank test, P = 0.007), or IIYGPAYSGR (PSG4) ($\geq$ 0.199; log-rank test, P = 0.004) who under-went rescue cerclage for cervical insufficiency, exhibited significantly shorter cerclage-to-deliv-ery intervals (Fig 4). Low plasma FAVEEIIQK (LXN) levels ($\leq$ 0.512) displayed an almost significant association with shorter cerclage-to-delivery intervals (log-rank test, P = 0.053). Likewise, the Cox proportional hazards model indicated that high plasma levels of DVLTFTCEPK (PSG4) and IIYGPAYSGR (PSG4), but not LEGEACGVYTPR (IGFBP-2) or FAVEEIIQK (LXN), were significantly associated with shorter cerclage-to-delivery intervals, after adjusting for advanced cervical dilatation ($\geq$ 3cm) (Table 5).

## Discussion

The main findings of this study were as follows: (1) utilizing TMT-based quantitative proteo-mic analysis, 40 DEPs were characterized in pooled plasma samples of women who experi-enced SPTD prior to 33 weeks of gestation after rescue cerclage; (2) among these 40 DEPs

**Table 4. Regression coefficients, ORs, and 95% CIs of the best protein panel*** for predicting spontaneous preterm delivery (SPTD) at <33 weeks of gestation.

| Predictor | Beta-coefficient | SE | OR (95% CI) | P-value |
|---|---|---|---|---|
| High LEGEACGVYTPR (IGFBP-2) ($\geq 0.098$)[†] | 3.167 | 1.230 | 23.74 (2.13–264.42) | 0.01 |
| High DVLTFTCEPK (PSG4) ($\geq 0.035$)[†] | 2.523 | 1.226 | 12.47 (1.13–137.71) | 0.04 |
| Low FAVEEIIQK (LXN) ($\leq 0.512$)[†] | 2.29 | 1.011 | 9.92 (1.37–71.89) | 0.023 |
| Constant | -3.836 | 1.415 | 0.022 | 0.007 |

SE, standard error; OR, odds ratio; CI, confidence interval; IGFBP-2, insulin-like growth factor-binding protein 2; PSG4, pregnancy specific beta-1-glycoprotein 4; LXN, latexin.

[†]Variables were dichotomized: high LEGEACGVYTPR ($\geq 0.098$ vs. $< 0.098$), high DVLTFTCEPK ($\geq 0.035$ vs. $< 0.035$), and low FAVEEIIQK ($\leq 0.512$ vs. $> 0.512$).

*Formula that was generated to predict SPTD at $< 33$ weeks was as follows: $Y = \log_e(Z) = -3.836 + 3.167$ (if LEGEACGVYTPR was $\geq 0.098$) + 2.523 (if DVLTFTCEPK was $\geq 0.035$) + 2.29 (if FAVEEIIQK was $\leq 0.512$). $Z = e^y$ and risk (%) = $[Z/(1 + Z)] \times 100$.

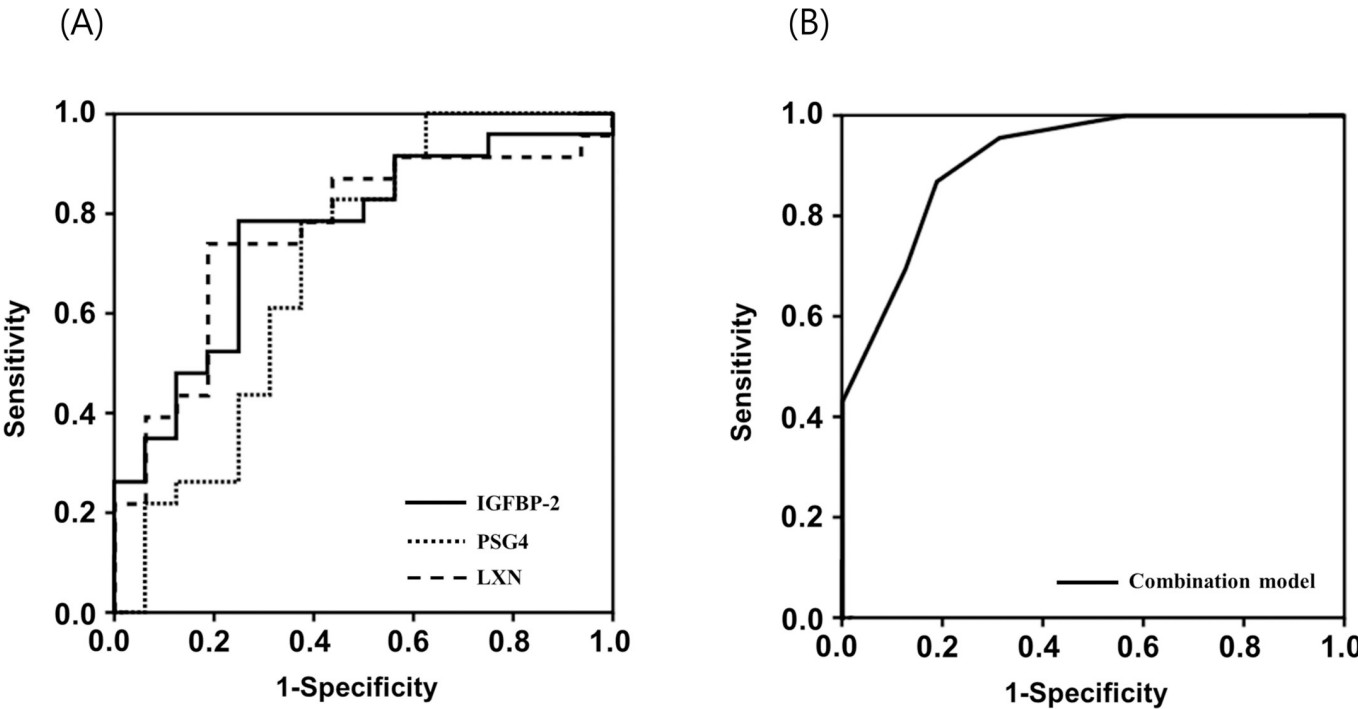

**Fig 3.** (A) Receiver operating characteristic (ROC) curves for plasma LEGEACGVYTPR (IGFBP-2), DVLTFTCEPK (PSG4), and FAVEEIIQK (LXN) levels in predicting spontaneous preterm delivery (SPTD) at < 33 weeks of gestation after cerclage placement (LEGEACGVYTPR: area under the curve [AUC] = 0.758, standard error [SE] = 0.079; DVLTFTCEPK: AUC = 0.696, SE = 0.095; and FAVEEIIQK: AUC = 0.764, SE = 0.080). (B) ROC curve for the best combined predictive model (including plasma LEGEACGVYTPR [IGFBP-2], DVLTFTCEPK [PSG4], and FAVEEIIQK [LXN]) for predicting SPTD at < 33 weeks of gestation. The AUC for the combined predictive model was 0.905 ($P < 0.05$ for LEGEACGVYTPR vs the combined predictive model, $P < 0.05$ for DVLTFTCEPK vs the combined predictive model, and $P < 0.05$ for plasma FAVEEIIQK vs the combined predictive model).

validated by MRM-MS analysis, three plasma proteins (IGFBP-2, LEGEACGVYTPR; PSG4, DVLTFTCEPK and IIYGPAYSGR; and LXN, FAVEEIIQK) were confirmed as potential biomarkers for pregnancy outcome following rescue cerclage in women with cervical insufficiency, independent of well-known risk factors, such as cervical dilatation; and (3) a multibiomarker panel (with an AUC of 0.916), consisting of LEGEACGVYTPR (IGFBP-2), DVLTFTCEPK (PSG4), and FAVEEIIQK (LXN) was more closely associated with SPTD at < 33 weeks of gestation after cerclage, than individual biomarker levels. To the best of our knowledge, this is the first proteomic study of plasma samples to identify biomarkers associated with poor pregnancy outcomes in women with cervical insufficiency undergoing rescue cerclage. The biomolecules identified in the current study may contribute to a better understanding of the biochemical mechanisms responsible for SPTD after rescue cerclage and they represent targets for the development of novel therapeutics.

In the current study, the AUC values for IGFBP-2, PSG4, and LXN ranged from 0.686 to 0.764, when used as single markers to predict cerclage failure following rescue cerclage. However, their overall diagnostic performance as single markers was not sufficient to be used in clinical practice. Utilizing a stepwise regression model, a combined multi-biomarker panel comprising 3 plasma proteins (IGFBP-2, PSG4, and LXN) greatly improved the predictive accuracy to an AUC of 0.916, demonstrating that this multi-biomarker panel had a significantly better overall diagnostic performance than each of the biomarkers alone. These observations are in line with the results of previous studies on disorders with complex pathogeneses (e.g., preterm birth and preeclampsia) [20–22] and suggested that the molecular mechanism

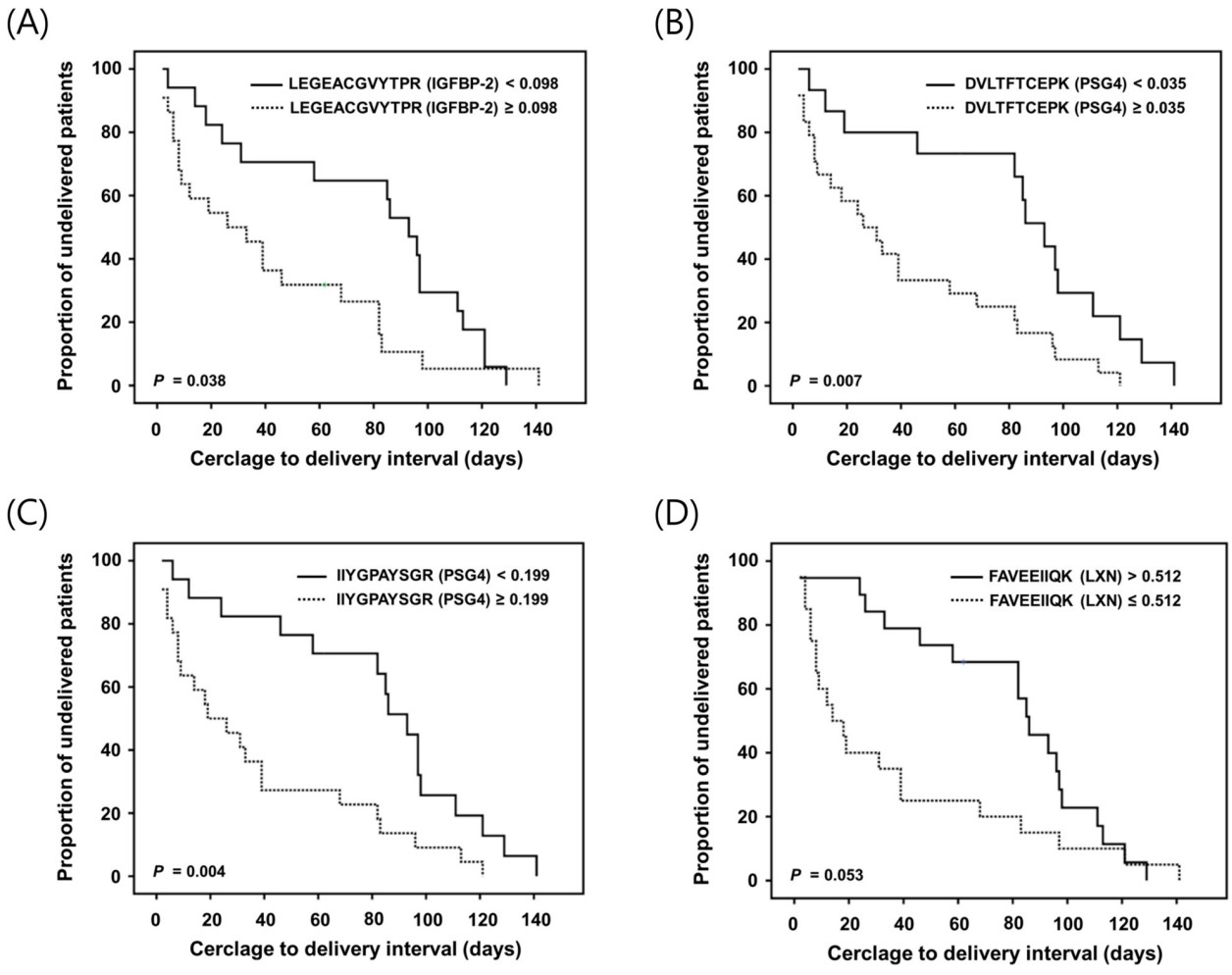

**Fig 4.** Kaplan-Meier survival estimates of the cerclage-to-delivery interval for (A) LEGEACGVYTPR (IGFBP-2) of ≥0.098 or <0.098 (median, 26.00 days [95% CI, 1.17–50.82] vs. 93.00 days [95% CI, 78.21–107.79]; $P$ = 0.038), (B) DVLTFTCEPK (PSG4) of ≥0.035 or <0.035 (median, 26.00 days [95% CI, 7.99–44.00] vs. 93.00 days [95% CI, 78.81–107.18]; $P$ = 0.007), (C) IIYGPAYSGR (PSG4) of ≥0.199 or <0.199 (median, 19.00 days [95% CI, 0.00–38.53] vs. 93.00 days [95% CI, 77.82–108.17]; $P$ = 0.004), and (D) FAVEEIIQK (LXN) of ≤0.512 or >0.512 (median, 14.00 days [95% CI, 0.85–27.14] vs. 86.00 days [95% CI, 71.23–100.76]; $P$ = 0.053). IGFBP, insulin-like growth factor-binding protein; CI, confidence interval; PSG, pregnancy-specific beta 1 glycoprotein; LXN, latexin.

**Table 5. Cox proportional hazards analysis of cerclage-to-delivery interval.**

| Variable[a] | Adjusted hazard ratio (95% confidence interval)[b] | $P$-value[c] |
|---|---|---|
| High DVLTFTCEPK (PSG4) (≥ 0.035) | 2.87 (1.38–5.99) | 0.005 |
| High IIYGPAYSGR (PSG4) (≥ 0.199) | 2.64 (1.31–5.34) | 0.007 |
| High LEGEACGVYTPR (IGFBP-2) (≥ 0.098) | 1.91 (0.95–3.86) | 0.070 |
| Low FAVEEIIQK (LXN) (≤ 0.512) | 1.93 (0.99–3.77) | 0.052 |

[a] Variables were dichotomized: high DVLTFTCEPK (≥ 0.035 vs. < 0.035), high IIYGPAYSGR (≥ 0.199 vs. < 0.199), high LEGEACGVYTPR (≥ 0.098 vs. < 0.098), and low FAVEEIIQK (≤ 0.512 vs. > 0.512).

[b] Adjusted for cervical dilatation ≥ 3cm.

[c] For the adjusted hazard ratio.

for the development of SPTD after rescue cerclage for cervical insufficiency is multifactorial. Finally, the clinical relevance of our findings is that plasma protein biomarkers (assessed by MRM-MS quantitative proteomics) in patients undergoing rescue cerclage for cervical insufficiency could be beneficial for identification of adverse pregnancy outcomes. This information may assist clinicians (at least in part) in non-invasively selecting optimal candidates for rescue cerclage, and providing personalized counseling based on patient-specific risks.

**An** important finding from the present study that should be highlighted is that the plasma levels of IGFBP-2, PSG4, and LXN were independent predictors of the outcome of rescue cerclage in women with cervical insufficiency and they were assayed in a non-invasive manner. IGFBP-2 is known to regulate the activity of insulin growth factor, which is involved in the control of placental and fetal growth and development [23] and is expressed in human fetal and placental tissues [24, 25]. Previous studies have shown that serum IGFBP-2 levels are not significantly altered by preterm birth, whereas serum IGFBP-1 levels are significantly altered [26, 27]. In contrast, our group previously reported that low plasma levels of IGFBP-2 were independently associated with histologic chorioamnionitis in women with preterm labor [28]. However, in the present study, IGFBP-2 levels were significantly increased in plasma from patients with poor pregnancy outcome after rescue cerclage. This is in accordance with a previous proteomic study using plasma from asymptomatic women at 17–28 weeks of gestation, in which IGFBP-2 was detected as a protein that distinguished cases of preterm birth from term controls [29]. We cannot explain the discrepancy between the present findings and those of earlier reports on preterm labor [26, 27], but the clinical significance and mechanisms of action of IGFBP-2 identified in the present study require further confirmation in large cohort studies. Finally, in the current study, the plausible mechanisms by which increased plasma IGFBP-2 levels in cervical insufficiency contribute to SPTD (cerclage failure) may be related to abnormal placental function, leading to the activation of placental vascular insufficiency, a known significant risk factor associated with SPTD [30]. This notion is supported by previous studies showing that IGFBP-2 and its family of proteins have a significant association with placental dysfunction, particularly due to maternal vascular insult [31, 32].

PSG4 is a member of a family of proteins that are synthesized by placental trophoblasts and released into the maternal blood circulation throughout pregnancy [33, 34]. Maternal plasma levels of PSG4 are significantly altered in cases of adverse pregnancy outcome (e.g., fetal growth retardation, pre-eclampsia, and preterm birth) compared to normal pregnancies [34, 35]. Furthermore, the *PSG4* gene is associated with the molecular mechanisms of smoking-induced placental abnormalities [36]. Thus, we speculated that elevated levels of plasma PSG4 in women with cerclage failure may reflect abnormal placentation and the activation of placental vascular insufficiency, and thus they may be associated with an increased risk of SPTD after cerclage.

Latexin is the only known mammalian carboxypeptidase inhibitor. It may play a role in inflammation and innate immune pathways [37] and it has been reported to be a putative tumor suppressor protein in several studies showing the significant down-regulation of its mRNA and proteins levels in several cancer types [38–40]. However, to date, there has been no report on the levels of latexin in the plasma of women with preterm birth-related disorders, including cervical insufficiency and preterm labor, or on its role during pregnancy. In the present study, lower latexin levels were significantly and independently associated with SPTD at < 33 weeks following rescue cerclage. Given that inflammation is a critical component of tumor progression and the tumor microenvironment is mostly regulated by inflammatory cells [41], our results regarding latexin are supported by several reports showing that inflammation is associated with an increased risk of SPTD after emergency cerclage [6].

There are several limitations of this study. First, the current study included a small number of study subjects from a single center, the study was retrospective, and the validation of

candidate proteins from our discovery experiments was not performed in a completely independent set of samples. These factors may limit the generalizability of our results and thus, these findings should be replicated in a larger, independent cohort. Second, our proteomic analysis for the discovery of biomarker candidates was performed on pooled samples, despite the fact that sample pooling in proteomic approaches may not closely represent the biological average of the individual samples [42]. In fact, Molinari et al. have reported that pooling of samples reduces the power of statistical tests, resulting in a reduced number of differential peaks and an increase in the detection of false differential peaks, when compared to using individual samples [42]. Third, this study was performed in only one ethnic group (a Korean ethnic group). Given previous reports that biomarkers of SPTD differ in different races [43], our proteomics results should be replicated in women of other ethnic backgrounds, in a larger cohort. Fourth, the MRM results were not further validated by ELISA, limiting their application in routine clinical practice. The strengths of our study are as follows: (1) this is the first comprehensive assessment of plasma proteomic profiles associated with poor outcome after rescue cerclage; (2) several novel potential biomarkers of the clinical outcome of cerclage were identified in a complex biological sample; and (3) we included an adjustment for the effect of cervical dilation (which is considered an important clinical factor in this context) in our multivariate analysis. In the present study, SPTD prior to 33 weeks after rescue cerclage was selected as the primary outcome measure, as the incidence of neonatal morbidity and mortality associated with prematurity decreased significantly in neonates born after this gestational age [44]. In addition, we performed targeted MRM-MS quantitative proteomics instead of ELISA because it has advantages of (1) cost-effectiveness, (2) ability to simultaneously quantify multiple peptides, and (3) generation of consistent, accurate, and reproducible datasets between laboratories [45–47].

## Conclusions

Our study demonstrated that a targeted proteomics-based approach can be useful for identification of biomarkers in blood samples that can potentially be used in clinical practice to predict the outcome of rescue cerclage. Plasma levels of IGFBP-2, PSG4, and LXN may be used as independent biomarkers to predict poor pregnancy outcome following rescue cerclage, with good accuracy, particularly when used as a combined multi-biomarker panel. Further studies are needed to investigate the possible mechanistic role of these biomarkers in promoting preterm birth after rescue cerclage for cervical insufficiency.

## Supporting information

**S1 Table. Demographic and clinical characteristics of women in the exploratory cohort.**
(XLSX)

**S2 Table. List of identified proteins from control (cerclage success, n = 10) and case (cerclage failure, n = 10) groups.**
(XLSX)

**S3 Table. List of plasma proteins that exhibited significant differences in a pairwise comparison of spontaneous preterm delivery < 33 weeks vs. ≥ 33 weeks, after rescue cerclage in women with cervical insufficiency, using TMT-based quantitative proteomics.**
(XLSX)

**S4 Table. List of surrogate peptides for MRM method development.**
(XLSX)

**S5 Table. Optimized MRM method parameters of 404 transitions for 59 peptides from 37 proteins.**
(XLSX)

**S1 Fig. A heatmap with dendrogram of hierarchical cluster analysis of 40 differentially expressed proteins.** Case and control refer to plasma samples acquired from patients who had subsequent spontaneous preterm delivery at <33 weeks after cerclage placement (case) and who delivered at ≥33 weeks (control). (red = increased, green = decreased).
(TIF)

**S2 Fig. Gene-ontology enrichment analysis of the identified differentially expressed proteins as determined by the DAVID bioinformatics database tool.**
(TIF)

**S3 Fig. Representative response curves of heavy peptides.**
(TIF)

**S1 File. Raw data for the exploratory cohort.**
(SAV)

**S2 File. Raw data for the total cohort.**
(SAV)

**S3 File.**
(PDF)

# Acknowledgments

We are grateful to the patients in the study. We thank our medical staffs for their assistance.

# Author Contributions

**Conceptualization:** Kisoon Dan, Ji Eun Lee, Sun Min Kim, Kyo Hoon Park.

**Data curation:** Kisoon Dan, Ji Eun Lee, Sun Min Kim, Subeen Hong, Hyeon Ji Kim, Kyo Hoon Park.

**Formal analysis:** Kisoon Dan, Ji Eun Lee, Dohyun Han, Subeen Hong, Hyeon Ji Kim, Kyo Hoon Park.

**Funding acquisition:** Kyo Hoon Park.

**Investigation:** Dohyun Han, Sun Min Kim, Subeen Hong, Hyeon Ji Kim, Kyo Hoon Park.

**Methodology:** Kisoon Dan, Ji Eun Lee, Dohyun Han, Sun Min Kim, Subeen Hong, Hyeon Ji Kim, Kyo Hoon Park.

**Project administration:** Kyo Hoon Park.

**Supervision:** Dohyun Han, Kyo Hoon Park.

**Validation:** Kisoon Dan, Kyo Hoon Park.

**Writing – original draft:** Kisoon Dan, Ji Eun Lee, Sun Min Kim, Kyo Hoon Park.

**Writing – review & editing:** Kisoon Dan, Ji Eun Lee, Dohyun Han, Sun Min Kim, Subeen Hong, Hyeon Ji Kim, Kyo Hoon Park.

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
