## [Decision Letter · Decision Letter 0]

24 Feb 2021

PONE-D-21-00572

Proteomic identification of biomarkers in maternal plasma that predict the outcome of rescue cerclage for cervical insufficiency

PLOS ONE

Dear Dr. Park,

Thank you for submitting your manuscript to PLOS ONE. After careful consideration, we feel that it has merit but does not fully meet PLOS ONE’s publication criteria as it currently stands. Therefore, we invite you to submit a revised version of the manuscript that addresses the points raised during the review process.

Three experts in the field handled your manuscript, and we are appreciative of their time and efforts. Although interest was found in your study, several comments arose that require your attention. Please address ALL of the reviewers' comments in your revised manuscript.

We look forward to receiving your revised manuscript.

Kind regards,

Frank T. Spradley

Academic Editor

PLOS ONE

2. We noticed you have some minor occurrence of overlapping text with the following previous publications, which needs to be addressed:

- https://link.springer.com/article/10.1007/s43032-019-00110-8

- https://iovs.arvojournals.org/Article.aspx?articleid=2751401

- https://jkms.org/DOIx.php?id=10.3346%2Fjkms.2020.35.e26

- https://obgyn.pericles-prod.literatumonline.com/doi/10.1111/jog.12976

In your revision ensure you cite all your sources (including your own works), and quote or rephrase any duplicated text outside the methods section. Further consideration is dependent on these concerns being addressed.

Reviewers' comments:

Reviewer's Responses to Questions

**Comments to the Author**

1. Is the manuscript technically sound, and do the data support the conclusions?

Reviewer #1: Yes

Reviewer #2: Yes

Reviewer #3: Partly

2. Has the statistical analysis been performed appropriately and rigorously? 

Reviewer #1: Yes

Reviewer #2: Yes

Reviewer #3: I Don't Know

3. Have the authors made all data underlying the findings in their manuscript fully available?

Reviewer #1: Yes

Reviewer #2: Yes

Reviewer #3: Yes

4. Is the manuscript presented in an intelligible fashion and written in standard English?

Reviewer #1: Yes

Reviewer #2: Yes

Reviewer #3: No

5. Review Comments to the Author

Reviewer #1: I have read the paper entitled “Proteomic indentification of biomarkes in maternal plasma that predict the outcome of rescue cerclage for cervical insufficiency” written by Kisoon Dan.

The authors have performed two-stage experiment. In the discovery phase that employed liquid chromatrography-tandem mass spectrometry to identify the difference in plasma proteome composition between the samples obtained from pregnant women with cervical insufficiency with bulging membranes with and without favorable outcomes. In the verification phase, they use MRM-MS to verify the difference in the plasma levels of targeted proteins between women who underwent cerclage with and without favorable outcomes .

The papers seems to be methodologically correct and it is well written and easy to follow.

1) It is obvious that unfavorable outcome in women with cervical insufficiency with prolapsed membranes is in those who had intra-amniotic infection or sterile intra-amniotic inflammation. Did you see any differences in maternal inflammatory mediators such CRP, IL-6, WBC between the groups?

2) Why did you use “a semi-pooling” strategy (3 pools with 3, 3, and 4 samples) in the discovery phase instead of a pooling strategy or individual sample analyses?

3) The use of MRM-MS is quite far from a clinical setting. Why did you decide to employ this proteomic instrument in the verification phase instead of antibody based approach?

4) Are the commercial ELISA kits available for the targeted proteins?

5) What is a clinical relevance of your results?

Reviewer #2: The article entitled “Proteomic identification of biomarkers in maternal plasma that predict the outcome of rescue cerclage for cervical insufficiency” is about an interesting topic and the data were analyzed properly and written well.

Reviewer #3: Interesting approach to evaluate new biomarkers for prediction of success of rescue cerclage.

Line 75-77:

What do the authors mean by “select optimal candidates for rescue cerclage and thus,reduce the risk…”

Would you recommend to rely on biomarkers for the decision to opt for a rescue cerclage?

Line 109: What was your control group?

Did these patients also receive a rescue cerclage, or a prophylactic cerclage or no cerclage at all?

What was the time interval in your study cohort (rescue cerclage patients) between placement of cerclage and PTD? How many weeks could the pregnancy be prolonged by the rescue procedure?

I think an interesting end point would rather be the prolongation of pregnancy (in weeks) and not the cut-off of delivery before (completed?) 33 weeks.

How are the biomarkers in patients with early cerclage failure compared to those that carried to 32 weeks?

As the authors already acknowledge in the limitations, samples were not collected prospectively.

There are several run-on sentences throughout the manuscript that are quite difficult to follow.

6. PLOS authors have the option to publish the peer review history of their article (what does this mean?). If published, this will include your full peer review and any attached files.

Reviewer #1: No

Reviewer #2: No

Reviewer #3: No

---

## [Author Response · Author response to Decision Letter 0]

18 Mar 2021

Manuscript ID: PONE-D-21-00572

March 17, 2021

RE: Manuscript ID: PONE-D-21-00572

“Proteomic identification of biomarkers in maternal plasma that predict the outcome of rescue cerclage for cervical insufficiency” 

Dear Editor:

Thank you very much for the review of our manuscript. The comments of the reviewers were constructive and have been used to revise and improve the manuscript.

The following is an itemized account of the changes in the manuscript made in response to comments.

Response to the Editor

Point #1: The Editor made the following comments for us:

Response: According to the editor’s suggestion, we recheck our manuscript and ensure that our paper meets PLOS ONE's style requirements, including those for file naming.

Point #2: The Editor made the following comments for us:

We noticed you have some minor occurrence of overlapping text with the following previous publications, which needs to be addressed:

- https://link.springer.com/article/10.1007/s43032-019-00110-8

- https://iovs.arvojournals.org/Article.aspx?articleid=2751401

- https://jkms.org/DOIx.php?id=10.3346%2Fjkms.2020.35.e26

- https://obgyn.pericles-prod.literatumonline.com/doi/10.1111/jog.12976

In your revision ensure you cite all your sources (including your own works), and quote or rephrase any duplicated text outside the methods section. Further consideration is dependent on these concerns being addressed.

Response: We also check plagiarism using iThenticate (also known as CrossCheck or Similarity Check), and do our best to rephrase or quote any duplicated text outside the methods section. We highlight rephrased sentence in the manuscript. 

Response to the Reviewer #1

Point #1: The reviewer asked us to see any differences in maternal inflammatory mediators, such as CRP, IL-6, and WBC between the groups because it is obvious that unfavorable outcome in women with cervical insufficiency with prolapsed membranes is in those who had intra-amniotic infection or sterile intra-amniotic inflammation. 

Response: Good point. We have blood CRP and WBC data for the 39 participants included in the study. However, we did not measure the plasma IL-6 levels using ELISA, thereby being unavailable for plasma IL-6 data. Thus, as the reviewer suggested, we analyze the data of blood CRP levels and WBC counts based on the occurrence of SPTD at < 33 weeks after cerclage, and the results are as follows: 

 Delivery at <33 weeks (n = 23) Delivery at ≥33 weeks (n = 16) P-value

Serum C-reactive protein (mg/L) 3.3 (0.5 – 33.9) 5.2 (0.1 – 32.0) 0.361

White blood cells count (×103/mm3) 11.0 (6.5 – 18.6) 10.5 (7.1 – 13.3) 0.278

As shown in the above Table, the median serum CRP levels and WBC counts did not significantly differ between clinical success (delivery at ≥33 weeks) and failure (delivery at <33 weeks) groups. We add these results to the Table 2.

According to the addition of “CRP levels and WBC counts”, we add the following sentences to the 4th paragraph of the Materials and Methods section (in line 131, page 7).

White blood cell counts and C-reactive protein concentrations in the maternal blood samples were measured.

Point #2: The reviewer asked us why you used “a semi-pooling” strategy (3 pools with 3, 3, and 4 samples) in the discovery phase instead of a pooling strategy or individual sample analyses. 

Response: Excellent point. As reviewer knows, the advantages of sample pooling included (1) the possibility of reducing individual proteome variations not related to the disease (that is, highlighting the most consistent disease related alterations); (2) an important reduction in the amount of protein required from each sample, allowing experimental replicates and subsequent studies; and (3) a significant reduction of the costs per test. However, the disadvantage of sample pooling is the inability to collect information on individual variation, as we described in the 6th paragraph of the Discussion section as the limitation of the study. Moreover, in the discovery experiments of the current study, we performed six-plex Tandem mass tag (TMT) labeling-based quantitative proteomics, allowing to compare protein (and peptide) amounts between six conditions. Thus, we thought that “a semi-pooling” strategy (3 pools with 3, 3, and 4 samples) has merit in terms of (1) reducing the disadvantage inherent in pooling all 10 samples from each group together, and (2) providing three independent biological replicates in each group that can be used for six-plex TMT labeling-based quantitative analyses.

We add the following sentences to the 5th paragraph of the Materials and Methods section (in line 138, page 8).

This pooling strategy (three pools, containing three, three, or four samples each) has merit in terms of (1) reducing the disadvantage inherent in pooling all 10 samples from each group together, and (2) providing three independent biological replicates in each group that can be used for six-plex TMT labeling-based quantitative analyses.

Point #3: The reviewer asked us why we decided to employ this proteomic instrument in the verification phase instead of antibody based approach, as the use of MRM-MS is quite far from a clinical setting.

Response: Excellent point. As reviewer knows, traditionally, antibody-based approach (i.e., ELISA) has been the most widely used technique for protein quantification. Moreover, as the reviewer mentioned, currently, ELISA methods have been routinely applied in clinical practice due to its advantages of speed and compatibility with standard clinical laboratory equipment. However, ELISA has major disadvantages, such as its technical limitations regarding multiplex quantitation and its cost and time-consuming development of specific antibodies. In contrast, multiple reaction monitoring-mass spectrometry (MRM-MS) is a targeted proteomic technique that does not require antibody and enables quantitative measurements of at least 100 protein targets per sample. In addition, MRM-MS has been known to generate consistent, accurate, and reproducible datasets between laboratories, even in highly complex samples. 

 In the current study, we attempted to verify 40 selected target proteins [(40 differentially expressed proteins (DEPs)]. Thus, for the verification of these 40 proteins, we employed the MRM-MS-based assays that are cost‐effective and suitable for multiplex quantitation of hundreds of proteins. To clarify this issue, we add the following sentences to the 6th paragraph of the Discussion section.

In addition, we performed targeted MRM-MS quantitative proteomics instead of ELISA because it has advantages of (1) cost-effectiveness, (2) ability to simultaneously quantify multiple peptides, and (3) generation of consistent, accurate, and reproducible datasets between laboratories [45-47].

We add the following references to the Reference section with adding the reason why we employed this proteomic instrument in the verification phase instead of antibody based approach. 

45. Gillette MA, Carr SA. Quantitative analysis of peptides and proteins in biomedicine by targeted mass spectrometry. Nat Methods. 2013;10(1):28-34. Epub 2012/12/28. doi: 10.1038/nmeth.2309. PubMed PMID: 23269374; PubMed Central PMCID: PMCPMC3943160.

46. Whiteaker JR, Lin C, Kennedy J, Hou L, Trute M, Sokal I, et al. A targeted proteomics-based pipeline for verification of biomarkers in plasma. Nat Biotechnol. 2011;29(7):625-34. Epub 2011/06/21. doi: 10.1038/nbt.1900. PubMed PMID: 21685906; PubMed Central PMCID: PMCPMC3232032.

47. Kennedy JJ, Abbatiello SE, Kim K, Yan P, Whiteaker JR, Lin C, et al. Demonstrating the feasibility of large-scale development of standardized assays to quantify human proteins. Nat Methods. 2014;11(2):149-55. Epub 2013/12/10. doi: 10.1038/nmeth.2763. PubMed PMID: 24317253; PubMed Central PMCID: PMCPMC3922286.

Point #4: The reviewer asked us if the commercial ELISA kits are available for the targeted proteins.

Response: As described in the 4th paragraph of the Results section, we verified 40 target proteins (40 DEPs) identified by the TMT-based quantitative analysis (Table S3). I am very sorry to say that, I believe, it is not necessary to check whether the commercial ELISA kits are available for all of the 40 targeted proteins found in the current study. However, we confirmed that the commercial ELISA kits were available for the 5 proteins (IGFBP-2, PSG4, PGLYRP2, MET, and LXN) showing a significant association with cerclage outcome using the MRM-MS experiment. Nevertheless, we further cannot measure plasma levels of these 5 proteins in the 39 participants included in the study due to a limitation of funding related to the current study and depletion of volume of plasma required to conduct this assay. We add the following sentences to the last paragraph of the Discussion section as the limitation of the study.

Fourth, the MRM results were not further validated by ELISA, limiting their application in routine clinical practice.

Point #5: The reviewer asked us what a clinical relevance of our results is. 

Response: Excellent point. In the clinical setting of acute cervical insufficiency, the non-invasive identification of patients likely to be benefit from rescue cercalge may help clinicians in selecting optimal candidates for emergency cerclage and providing personalized counseling based on the patients specific risks. However, currently, little information is available on whether protein biomarkers using plasma are beneficial for the non-invasive identification of adverse outcomes in patients undergoing emergency cerclage for cervical insufficiency, particularly when assessed using a high-throughput screening platform. Our study demonstrated that a targeted proteomics-based approach may be useful for identification of biomarkers in blood samples that can potentially be used in clinical practice to predict the outcome of rescue cerclage. In particular, plasma levels of IGFBP-2, PSG4, and LXN may be used as independent biomarkers to predict poor pregnancy outcome following rescue cerclage, with good accuracy, particularly when used as a combined multi-biomarker panel.

Regarding the clinical relevance of our results, we add the following sentences to the 2nd paragraph of the Discussion section.

Finally, the clinical relevance of our findings is that plasma protein biomarkers (assessed by MRM-MS quantitative proteomics) in patients undergoing rescue cerclage for cervical insufficiency could be beneficial for identification of adverse pregnancy outcomes. This information may assist clinicians (at least in part) in non-invasively selecting optimal candidates for rescue cerclage, and providing personalized counseling based on patient-specific risks.

 Response to the Reviewer #2

Point #1: The reviewer made the following comments for us.

The article entitled “Proteomic identification of biomarkers in maternal plasma that predict the outcome of rescue cerclage for cervical insufficiency” is about an interesting topic and the data were analyzed properly and written well.

Response: Many thanks.

Response to the Reviewer #3

Point #1: The reviewer made the following comments for us.

Line 75-77:

What do the authors mean by “select optimal candidates for rescue cerclage and thus, reduce the risk…” Would you recommend to rely on biomarkers for the decision to opt for a rescue cerclage?

Response: Excellent point. We agree with the reviewer that clinicians cannot recommend relying on biomarkers for the decision to opt for a rescue cerclage, as described in the 1st paragraph of the Introduction section. That is, an emergency (or rescue) cerclage is currently the only method to prolong pregnancy and salvage the fetus in women presenting with a dilated cervix and/or prolapsed membranes in the second trimester. Thus, when acute cervical insufficiency is found, cervical cerclage sergury should be considered. We delete the last sentence of the 2nd paragraph in the Introduction section, which is the following sentence: 

Such information is clinically relevant, because blood biomarkers can be used to easily and non-invasively select optimal candidates for rescue cerclage and thus, reduce the risk of the complications posed by emergency cerclage (e.g., membrane rupture, preterm delivery). 

Point #2: The reviewer asked us what our control group was (in line 109) and if these patients also received a rescue cerclage, or a prophylactic cerclage or no cerclage at all.

Response: As described in the 2nd paragraph of the Materials and methods section, and the Methods section of the Abstract, our control group for discovery set consisted of 10 patients who delivered at ≥ 33 weeks after a rescue cerclage placement for cervical insufficiency. To clarify this issue, we add “who had rescue cerclage for cervical insufficiency” to the 3rd sentence of the 2nd paragraph in the Materials and methods section. The new 3rd sentence of the 2nd paragraph in the Materials and methods section now reads:

Each control patient who had rescue cerclage for cervical insufficiency was matched for gestational age at sampling, cervical dilatation, parity, years of cerclage placement, and maternal age with a case patient.

Point #3: The reviewer made the following comments for us.

What was the time interval in your study cohort (rescue cerclage patients) between placement of cerclage and PTD? How many weeks could the pregnancy be prolonged by the rescue procedure?

Response: In the current study, the mean cerclage-to-delivery interval was 55.72 ± 43.74 days (range, 2–141 days). We add the following sentence to the 6th paragraph of the Results section. 

The mean cerclage-to-delivery interval was 55.72 ± 43.74 days (range, 2–141 days).

Point #4: The reviewer made the following comments for us.

I think an interesting end point would rather be the prolongation of pregnancy (in weeks) and not the cut-off of delivery before (completed?) 33 weeks.

Response: Excellent point. According to the reviewer’s suggestion, we reanalyzed the data based on the cerclage-to-delivery interval, which was estimated using Kaplan-Meier analysis with the log-rank test and Cox's regression. We add the following survival analyses to the last paragraph of the Results section: 

Plasma protein markers and the cerclage-to-delivery interval

Kaplan-Meier survival analyses showed that patients with higher plasma levels of LEGEACGVYTPR (IGFBP-2) (≥0.098; log-rank test, P = 0.038), DVLTFTCEPK (PSG4) (≥ 0.035; log-rank test, P = 0.007), or IIYGPAYSGR (PSG4) (≥ 0.199; log-rank test, P = 0.004) who underwent rescue cerclage for cervical insufficiency, exhibited significantly shorter cerclage-to-delivery intervals (Figure 4). Low plasma FAVEEIIQK (LXN) levels (≤ 0.512) displayed an almost significant association with shorter cerclage-to-delivery intervals (log-rank test, P = 0.053). Likewise, the Cox proportional hazards model indicated that high plasma levels of DVLTFTCEPK (PSG4) and IIYGPAYSGR (PSG4), but not LEGEACGVYTPR (IGFBP-2) or FAVEEIIQK (LXN), were significantly associated with shorter cerclage-to-delivery intervals, after adjusting for advanced cervical dilatation (≥ 3cm) (Table 5). 

Table 5. Cox proportional hazards analysis of cerclage-to-delivery interval 

Variablea Adjusted hazard ratio (95% confidence interval)b P-valuec

High DVLTFTCEPK (PSG4) (≥ 0.035) 2.87 (1.38–5.99) 0.005

High IIYGPAYSGR (PSG4) (≥ 0.199) 2.64 (1.31–5.34) 0.007

High LEGEACGVYTPR (IGFBP-2) (≥ 0.098) 1.91 (0.95–3.86) 0.070

Low FAVEEIIQK (LXN) 

(≤ 0.512) 1.93 (0.99–3.77) 0.052

a Variables were dichotomized: high DVLTFTCEPK (≥ 0.035 vs. < 0.035), high IIYGPAYSGR (≥ 0.199 vs. < 0.199), high LEGEACGVYTPR (≥ 0.098 vs. < 0.098), and low FAVEEIIQK (≤ 0.512 vs. > 0.512).

b Adjusted for cervical dilatation ≥ 3cm

c For the adjusted hazard ratio.

We add the following sentences to the Statistical analysis section to explain how to perform survival analysis on plasma markers.

A Kaplan-Meier survival curve was used to analyze the interval from cerclage to delivery, and log-rank tests were performed to evaluate differences in the cerclage-to-delivery interval between the two curves. The data were fitted to Cox proportional hazards models for multivariate analysis, adjusting for advanced cervical dilatation.

Kaplan-Meier estimates of cerclage-to-delivery intervals for these four proteins in plasma are as follows and added as the Fig. 4 and legend of Fig. 4.

Fig. 4 Kaplan-Meier survival estimates of the cerclage-to-delivery interval for (A) LEGEACGVYTPR (IGFBP-2) of ≥0.098 or <0.098 (median, 26.00 days [95% CI, 1.17–50.82] vs. 93.00 days [95% CI, 78.21–107.79]; P = 0.038), (B) DVLTFTCEPK (PSG4) of ≥0.035 or <0.035 (median, 26.00 days [95% CI, 7.99–44.00] vs. 93.00 days [95% CI, 78.81–107.18]; P = 0.007), (C) IIYGPAYSGR (PSG4) of ≥0.199 or <0.199 (median, 19.00 days [95% CI, 0.00–38.53] vs. 93.00 days [95% CI, 77.82–108.17]; P =0.004), and (D) FAVEEIIQK (LXN) of ≤0.512 or >0.512 (median, 14.00 days [95% CI, 0.85–27.14] vs. 86.00 days [95% CI, 71.23–100.76]; P =0.053). IGFBP, insulin-like growth factor-binding protein; CI, confidence interval; PSG, pregnancy-specific beta 1 glycoprotein; LXN, latexin.

Point #5: The reviewer made the following comments for us.

How are the biomarkers in patients with early cerclage failure compared to those that carried to 32 weeks?

Response: According to the reviewer’s suggestion, we reanalyzed the data based on the occurrence of SPTD before or at 31 weeks and 6 days (<32 0/7 weeks of gestation; completed 32 weeks), and the results are as follows:

Table Characteristics of the study population in relation to the occurrence of spontaneous preterm delivery at < 32.0 weeks after cerclage.

 Delivery at <32.0 weeks (n = 21) Delivery at ≥32.0 weeks (n = 18) P-value

Age (years) 31.0 (27.0 – 36.0) 33.0 (24.0 – 39.0) 0.145

Nulliparity 61.9% (13/21) 50% (9/18) 0.455

Gestational age at sampling (weeks) 22.0 (17.3 – 25.1) 22.3 (20.0 – 25.4) 0.310

Cervical dilatation (cm) 3.0 (1.5 – 5.0) 1.5 (0.5 – 4.0) <0.001

≥3 cm 66.7% (14/21) 16.7% (3/18) 0.003

<3 cm 33.3% (7/21) 83.3% (15/18) 

Use of tocolytics 66.7% (14/21) 55.6% (10/18) 0.477

Use of corticosteroids 42.9% (9/21) 27.8% (5/18) 0.504

Use of antibiotics 100.0% (21/21) 100.0% (18/18) 

Gestational age at delivery (weeks) 24.6 (19.4 – 31.1) 36.6 (32.3 – 40.5) <0.001

LEGEACGVYTPR (IGFBP-2) 0.112 (0.065-0.231) 0.091 (0.070-0.153) 0.022

LIQGAPTIR (IGFBP-2) 0.083 (0.048-0.211) 0.068 (0.050-0.119) 0.078

DVLTFTCEPK (PSG4) 0.050 (0.013-0.285) 0.029 (0.006-0.304) 0.226

IIYGPAYSGR (PSG4) 0.314 (0.057-1.034) 0.151 (0.039-1.600) 0.135

TDCPGDALFDLLR (PGLYRP2) 0.980 (0.566-1.945) 0.844 (0.407-1.293) 0.108

GDLTIANLGTSEGR (MET) 0.037 (0.021-0.063) 0.042 (0.032-0.098) 0.076

FAVEEIIQK (LXN) 0.476 (0.208-0.938) 0.577 (0.323-0.754) 0.026

Values are given as median (range) or % (n/N).

As shown in the above table, the levels of LEGEACGVYTPR (IGFBP-2) and FAVEEIIQK (LXN) were significantly higher in women who had SPTD at < 32.0 weeks (≤ 31.6 weeks) than in women with SPTD at ≥ 32.0 weeks. However, the levels of LIQGAPTIR (IGFBP-2), DVLTFTCEPK (PSG4), IIYGPAYSGR (PSG4), and TDCPGDALFDLLR (PGLYRP2), and GDLTIANLGTSEGR (MET) did not significantly differ between women with SPTD at < 32.0 weeks and those at ≥ 32.0 weeks, although the levels of LIQGAPTIR (IGFBP-2), and GDLTIANLGTSEGR (MET) had borderline associations with the occurrence of SPTD at < 32.0 weeks after cerclage (P values = 0.078 and 0.076, respectively). These results are a little different from those of previous analyses in which SPTD at < 33.0 weeks (≤ 32.6 weeks) was used as an outcome measure, because the associations of DVLTFTCEPK (PSG4), IIYGPAYSGR (PSG4) with SPTD after cerclage disappear when using an outcome of SPTD at < 32.0 weeks instead of SPTD at < 33.0 weeks. 

 However, as shown in S1 File (Raw data for the exploratory cohort), from the exploratory experiment (a nested case-control design), we designed to study the role of plasma biomarkers as the outcome of SPTD at < 33.0 weeks (≤ 32.6 weeks), instead of SPTD at < 32.0 weeks (≤ 31.6 weeks). From the beginning of study design, we decided to use occurrence of SPTD before or at 32 6/7 weeks as the primary outcome measure (please see S1 File; the range for gestational age at delivery in case group was from 22 0/7 weeks to 32 3/7 weeks), and we believe that using this outcome may be appropriate in the study of rescue cerclage outcomes. 

Point #6: The reviewer made the following comments for us.

As the authors already acknowledge in the limitations, samples were not collected prospectively.

Response: As the reviewer mentioned, we already acknowledged that this study was designed retrospectively and described these limitations in the 6th paragraph of the Discussion section. 

Point #7: Reviewer said that there are several run-on sentences throughout the manuscript that are quite difficult to follow.

Response: Good point. We corrected several sentences in which the reviewer pointed out as run-on sentences, and highlighted these sentences in red. 

I hope that our revised manuscript and additional information that we provided in this letter meets with your approval. Please let us know if you have further questions or require additional information. I appreciate your time and assistance. 

With regards,

Kyo Hoon Park, MD, PhD

Professor

Department of Obstetrics and Gynecology. Seoul National University Bundang Hospital

166 Gumiro, Seongnamsi, Kyeonggido, 463-707, Korea

Tel: 82-31-787-7252

Fax: 82-31-787-4054

E-mail: pkh0419@snubh.org

Enclosures

---

## [Decision Letter · Decision Letter 1]

30 Mar 2021

Proteomic identification of biomarkers in maternal plasma that predict the outcome of rescue cerclage for cervical insufficiency

PONE-D-21-00572R1

Dear Dr. Park,

We’re pleased to inform you that your manuscript has been judged scientifically suitable for publication and will be formally accepted for publication once it meets all outstanding technical requirements.

Kind regards,

Frank T. Spradley

Academic Editor

PLOS ONE

Reviewers' comments:

Reviewer's Responses to Questions

**Comments to the Author**

1. If the authors have adequately addressed your comments raised in a previous round of review and you feel that this manuscript is now acceptable for publication, you may indicate that here to bypass the “Comments to the Author” section, enter your conflict of interest statement in the “Confidential to Editor” section, and submit your "Accept" recommendation.

Reviewer #1: All comments have been addressed

2. Is the manuscript technically sound, and do the data support the conclusions?

Reviewer #1: Yes

3. Has the statistical analysis been performed appropriately and rigorously? 

Reviewer #1: Yes

4. Have the authors made all data underlying the findings in their manuscript fully available?

Reviewer #1: Yes

5. Is the manuscript presented in an intelligible fashion and written in standard English?

Reviewer #1: Yes

6. Review Comments to the Author

Reviewer #1: All my comments have been addressed. I do not have any further questions or comments.

I recommend accepting the paper in its current form.

7. PLOS authors have the option to publish the peer review history of their article (what does this mean?). If published, this will include your full peer review and any attached files.

Reviewer #1: No

---

## [Editor Report · Acceptance letter]

5 Apr 2021

PONE-D-21-00572R1 

Proteomic identification of biomarkers in maternal plasma that predict the outcome of rescue cerclage for cervical insufficiency 

Dear Dr. Park:

I'm pleased to inform you that your manuscript has been deemed suitable for publication in PLOS ONE. Congratulations! Your manuscript is now with our production department. 

Kind regards, 

on behalf of

Dr. Frank T. Spradley 

Academic Editor

PLOS ONE